# A Novel Multi-Target Mu/Delta Opioid Receptor Agonist, HAGD, Produced Potent Peripheral Antinociception with Limited Side Effects in Mice and Minimal Impact on Human Sperm Motility In Vitro

**DOI:** 10.3390/molecules28010427

**Published:** 2023-01-03

**Authors:** Fangfang Li, Feng Yue, Wei Zhang, Biao Xu, Yiqing Wang, Xuehong Zhang

**Affiliations:** 1The First School of Clinical Medicine, Lanzhou University, Lanzhou 730000, China; 2Key Laboratory for Reproductive Medicine and Embryo of Gansu, Gansu International Scientific and Technological Cooperation Base of Reproductive Medicine Transformation Application, Lanzhou 730000, China; 3Key Laboratory of Preclinical Study for New Drugs of Gansu Province, School of Basic Medical Sciences, Lanzhou University, Lanzhou 730000, China

**Keywords:** pain, analgesic, opioid, infertility, sperm motility

## Abstract

Pain is a common clinical symptom among patients. Although various opioid analgesics have been developed, their side effects hinder their application. This study aimed to develop a novel opioid analgesic, HAGD (H-Tyr-D-AIa-GIy-Phe-NH_2_), with limited side effects. In vivo studies on mouse models as well as in vitro studies on Chinese hamster ovary (CHO) cells expressing human mu, delta, or kappa opioid receptors (CHO_hMOP_, CHO_hDOP_, and CHO_hKOP_, respectively) and human sperm were conducted. Compared with subcutaneous morphine (10 mg/kg), subcutaneous HAGD (10 mg/kg) produced equipotent or even greater antinociception with a prolonged duration by activating mu/delta opioid receptors in preclinical mouse pain models. The analgesic tolerance, rewarding effects (i.e., conditioned place preference and acute hyperlocomotion), and gastrointestinal transit inhibition of HAGD were significantly reduced compared with those of morphine. Both HAGD and morphine exhibited a withdrawal response and had no impacts on motor coordination. In CHO_hMOP_ and CHO_hDOP,_ HAGD showed specific and efficient intracellular Ca^2+^ stimulation. HAGD had minimal impact on human sperm motility in vitro, whereas 1 × 10^−7^ and 1 × 10^−8^ mol/L of morphine significantly declined sperm motility at 3.5 h. Overall, HAGD may serve as a promising antinociceptive compound.

## 1. Introduction

Pain is a common clinical symptom associated with most injuries and diseases and for which patients seek medical attention. Analgesics are divided into non-opioid (e.g., non-steroidal anti-inflammatory drugs, antidepressants, and anticonvulsants) and opioid (e.g., morphine, tramadol, and remifentanil) analgesics. Opioid analgesics are potent against moderate-to-severe acute and chronic pain [1]. They are typical mu opioid receptor (MOR) agonists dominated by morphine, fentanyl, and oxycodone [2,3]. However, the use of opioid analgesics is partly limited by their side effects, such as tolerance development, addiction, constipation, and sperm motility suppression [4,5,6,7]. According to previous studies, MOR activation may reduce sperm motility, whereas delta opioid receptor (DOR) activation may maintain sperm motility [6,7,8].

In 1997, Zadina et al. [9] discovered and isolated endomorphin-1 (Tyr1-Pro2-Trp3-Phe4-NH_2_) and endomorphin-2 (Tyr1-Pro2-Phe3-Phe4-NH_2_) from the bovine frontal cortex, which are highly selective for MORs and produce more potent and prolonged antinociception than morphine in mice. Soon after, Czapla et al. [10] elucidated that in rats, endomorphins have fewer effects on respiration and cardiovascular activity than morphine. In addition, the study demonstrated the inability of endomorphin-1 to cross the blood–brain barrier (BBB) into the central nervous system (CNS) as an analgesic [11]. Studies have shown that peripherally restricted MOR agonists can effectively modulate local inflammatory or neuropathic pain with reduced centrally mediated adverse effects [12]. In recent years, many studies have explored two opioid receptors simultaneously to develop drug candidates with a low risk of inducing tolerance [13]. Based on the abovementioned studies, our group designed a multi-target peptide, HAGD (H-Tyr-D-AIa-GIy-Phe-NH_2_). A radioligand binding assay and metabolic stability assessment have been performed in our previous study [14].

In this study, we investigated the antinociceptive effects of HAGD in a series of preclinical mouse pain models, including the radiant heat tail-flick test, carrageenan-induced inflammatory pain, formalin test, and acetic acid writhing. The side effects of HAGD were evaluated using tolerance, withdrawal response, conditioned place preference, acute hyperlocomotion, gastrointestinal transit (GIT), and rotarod tests. Chinese hamster ovary (CHO) cells expressing human mu, delta, or kappa opioid receptors (CHO_hMOP_, CHO_hDOP_, and CHO_hKOP,_ respectively) were used for calcium mobilization assays. In addition, the effect of HAGD on human sperm motility was evaluated in vitro.

## 2. Results

### 2.1. Calcium Mobilization Assays

The agonist efficacy of HAGD was examined at MORs, DORs, and KORs (Table 1). Intracellular Ca^2+^ stimulation was examined and compared with DAMGO (MOR agonist), DPDPE (DOR agonist), and CR845 (KOR agonist). In CHO_hMOP_ and CHO_hDOP_, HAGD showed similar potency to DAMGO or DPDPE (Figure 1A,B). The efficiency of HAGD was much lower than that of CR845 in CHO_hKOP_ cells (Figure 1C). HAGD is a specific and efficient agonist of MOR and DOR.

### 2.2. Antinociceptive Effects of Subcutaneous HAGD in the Tail-Flick Test

HAGD induced dose- and time-dependent increases in tail-flick latencies, which peaked at 15 min, whereas the tail-flick latency of morphine peaked at 30 min (Figure 2A). The antinociceptive ED_50_ value for HAGD was 2.84 mg/kg (equivalent to 6.24 μmol/kg) (95% CI, 1.98–3.54). HAGD (10 mg/kg) produced equipotent antinociception compared with morphine (10 mg/kg) in the AUC (Figure 2B). The ED_50_ for morphine analgesia was 5.24 mg/kg (equivalent to 16.28 μmol/kg) (95% CI, 4.57–5.95). Calcium mobilization assays indicated that HAGD activates MORs and DORs.

To further explore whether opioid receptors are involved in HAGD-induced antinociception, mice were pretreated with the classical opioid receptor antagonist naloxone and the peripherally restricted opioid receptor antagonist naloxone methiodide in the tail-flick test [15]. Subcutaneous pretreatment with naloxone (1 mg/kg equivalent to 2.5 μmol/kg, 10 min) (Figure 2C) or naloxone methiodide (1 mg/kg equivalent to 2.13 μmol/kg, 10 min) (Figure 2D) completely blocked the antinociceptive effect of HAGD. In contrast, s.c. HAGD-induced antinociception was not affected by i.c.v. injection of naloxone methiodide (5 nmol, 10 min) (Figure 2E). These results indicate that HAGD has poor BBB permeability. The figure (Figure 2) shows that the peripheral antinociception of HAGD was significantly blocked by the MOR and DOR antagonists β-funaltrexamine hydrochloride (1 mg/kg equivalent to 1.72 μmol/kg, 4 h) (Figure 2F) and naltrindole (1 mg/kg equivalent to 2.07 μmol/kg, 10 min) (Figure 2G), respectively, but not by the KOR antagonist nor-binaltorphimine dihydrochloride (1 mg/kg equivalent to 1.19 μmol/kg, 30 min) (Figure 2H).

### 2.3. Antinociceptive Effects of Subcutaneous HAGD in Carrageenan-Induced Inflammatory Pain

HAGD displayed dose- and time-dependent anti-allodynic activity in a carrageenan-induced inflammatory pain model. The anti-allodynic activity of HAGD and morphine lasted for 90 and 60 min, respectively, with a peak effect at 30 min post-injection (Figure 3A). Morphine (10 mg/kg equivalent to 31.07 μmol/kg) and HAGD (10 mg/kg equivalent to 21.98 μmol/kg) exhibited equipotent antinociception in the AUC (Figure 3B).

### 2.4. Antinociceptive Effects of Subcutaneous HAGD in Acetic Acid Writhing Test

The acetic acid writhing test, a widely used pain model, assesses visceral pain induced by irritant chemical stimulation. Intraperitoneal injection of 10 mL/kg of 0.6% acetic acid produced significant abdominal constrictions, and the average writhing number was 34 ± 2.49 in the saline group 5–15 min after intraperitoneal injection of acetic acid. HAGD produced a dose-dependent inhibition of the writhing response, and the average number with 1 mg/kg (equivalent to 2.2 μmol/kg) HAGD was 5.83 ± 1.94. Subcutaneous injection of 1 mg/kg (equivalent to 3.1 μmol/kg) morphine also inhibited the writhing response, with an average number of 4.60 ± 2.43. There was no significant difference between HAGD (1 mg/kg equivalent to 2.2 μmol/kg) and morphine (1 mg/kg equivalent to 3.1 μmol/kg) in the inhibition of the writhing response (Figure 3C).

### 2.5. Antinociceptive Effects of Subcutaneous HAGD in the Formalin Test

In the formalin test, intraplantar injection of 5% formalin resulted in a biphasic pain response (phase I: 0–5 min and phase II: 15–30 min) during the 30 min observation period. Phase I is induced by direct activation of nociceptors, whereas phase II represents the combined effects of nociceptor input and spinal cord sensitization. HAGD inhibited licking/flinching/biting behaviors induced by formalin in both phases I and II in a dose-dependent manner (Figure 3D). In addition, 10 mg/kg (equivalent to 31.07 μmol/kg) of morphine (s.c.) attenuated nociceptive responses in both phases I and II. Moreover, HAGD (10 mg/kg equivalent to 21.98 μmol/kg) had better analgesic effects than morphine (10 mg/kg equivalent to 31.07 μmol/kg) in phase I (Figure 3D).

### 2.6. Side Effects Evaluation of Subcutaneous HAGD

For antinociceptive tolerance, 10 mg/kg of HAGD induced non-tolerance antinociception for 8 consecutive days. The antinociceptive potency of morphine (10 mg/kg equivalent to 31.07 μmol/kg) decreased significantly from days 5 to 8 (Figure 4A). Both 10 mg/kg (equivalent to 21.98 μmol/kg) HAGD and 10 mg/kg (equivalent to 31.07 μmol/kg) morphine had a significant withdrawal response (Figure 4B). In the conditioned place preference, saline and 10 mg/kg (equivalent to 21.98 μmol/kg) of HAGD did not induce a place preference change; there was a significant place preference change with 10 mg/kg (equivalent to 31.07 μmol/kg) of morphine with conditioned place preference scores of 140.50 ± 43.96 compared with saline. Furthermore, there was a substantial difference in the conditioned place preference scores between the HAGD and morphine groups (Figure 4C). In the acute hyperlocomotion test, 10 mg/kg (equivalent to 21.98 μmol/kg) of HAGD did not show an apparent effect on locomotion compared with saline. In contrast, 10 mg/kg of morphine enhanced the locomotor activity of mice and significantly increased the total distance traveled compared with saline or HAGD (Figure 4D,E). In the GIT of mice, 10 mg/kg (equivalent to 21.98 μmol/kg) of HAGD induced a significant delay compared with saline. Morphine (10 mg/kg equivalent to 31.07 μmol/kg) also significantly weakened the GIT and exhibited a greater inhibitory effect than HAGD (Figure 4F). The potential influence of 10 mg/kg (equivalent to 21.98 μmol/kg) HAGD on motor function was evaluated using the rotarod test. Compared with the saline group, there was no significant difference in the endurance time of mice in the HAGD or morphine groups on the rotating rod (Figure 4G).

### 2.7. Effects on Sperm Motility of HAGD

Compared with the G-MOPS**^TM^** PLUS control group, the MOR/DOR agonist HAGD (0.0001, 0.001, 0.01, 0.1, 1, or 10 µM) did not produce significant changes in sperm PR during the 4-h incubation process (Figure 5A–F). The addition of the MOR agonist morphine (0.0001, 0.001, 1, or 10 µM) did not change the PR of sperm at 0.5, 1, 3.5, or 4 h compared with the control (Figure 5A,B,E,F). However, when spermatozoa were incubated with 0.01 or 0.1 µM of morphine for 3.5 h, the sperm PR was significantly reduced (45.75 ± 2.89 vs. 57.29 ± 2.22; 46.16 ± 1.15 vs. 57.29 ± 2.22, respectively; *p* < 0.05.) (Figure 5C,D).

## 3. Discussion

There is an urgent need for novel compounds that can provide opioid-like analgesia without side effects bothering doctors and patients. Our results showed that HAGD might be a promising compound for developing multi-target opioid analgesics with limited side effects.

The radioligand binding assay had been performed in our previous study to detect the MOR/DOR affinity (*Ki*) and selectivity of HAGD [14]. The *Ki*(MOR) and *Ki*(DOR) values of HAGD were found to be 2.9- and 1.5-fold lower, respectively, than those of endomorphin-1, suggesting that HAGD has better affinity and selectivity to MOR and DOR. In the tail-flick test, s.c. HAGD produced equipotent analgesic effects on acute heat irritation pain to morphine. Antinociception of HAGD was blocked by s.c. naloxone (1 mg/kg equivalent to 2.5 μmol/kg, 10 min) and s.c. naloxone methiodide (1 mg/kg equivalent to 2.13 μmol/kg, 10 min), but not by i.c.v. naloxone methiodide (5 nmol, 10 min), which cannot cross the BBB. The antinociceptive effect of s.c. morphine was not blocked by s.c. naloxone methiodide [16]. These findings demonstrate the poor ability of HAGD to penetrate the BBB into the CNS. Furthermore, the peripheral analgesic effects of HAGD were blocked by β-funaltrexamine hydrochloride (1 mg/kg equivalent to 1.72 μmol/kg, 4 h) and naltrindole (1 mg/kg equivalent to 2.07 μmol/kg, 10 min), respectively, demonstrating that HAGD exerted the effect of peripheral antinociception mediated via peripheral opioid receptors, MOR and DOR, consistent with the results of the radioligand binding and calcium mobilization assays. This mechanism may contribute to its reduced central side effects. In particular, following previous studies [16,17], the dose of the irreversible covalent opioid antagonist (β-funaltrexamine hydrochloride, 1 mg/kg equivalent to 1.72 μmol/kg, 4 h) and the reversible competitive antagonists (naloxone (1 mg/kg equivalent to 2.5 μmol/kg, 10 min), naloxone methiodide (1 mg/kg equivalent to 2.13 μmol/kg, 10 min), and naltrindole (1 mg/kg equivalent to 2.07 μmol/kg, 10 min)) in our study was 1 mg/kg, which was sufficient to produce antagonistic effects, whereas the dose of HAGD was 10 mg/kg (equivalent to 21.98 μmol/kg), which was nearly ten times higher than doses of these antagonists. In Table 1, the *Ki*(MOR) and *Ki*(DOR) values of naloxone were 1.1- and 1226-fold lower, respectively, than those of HAGD; the *Ki*(MOR) and *Ki*(DOR) values of naloxone methiodide were 0.1- and 3.8-fold lower, respectively, than those of HAGD. The *Ki*(MOR) value of β-funaltrexamine hydrochloride was 7.0-fold lower than that of HAGD. The *Ki*(DOR) value of naltrindole was 190,000-fold lower than that of HAGD. These results suggested that these antagonists (naloxone, naloxone methiodide, β-funaltrexamine hydrochloride, and naltrindole) had better affinity to MOR or DOR. The stronger affinity and earlier administration of these antagonists than those of HAGD may partly explain the sufficient antagonistic effects observed at 1 mg/kg and suggest that the low-dose competitive antagonists used in this study were able to reverse the analgesic effects of HAGD given in the dose (10 mg/kg equivalent to 21.98 μmol/kg) more than ten times higher than doses of antagonists. The metabolic stability of HAGD was tested in 15% mouse brain homogenate and 100% mouse serum [14]. The results showed that the half-life of HAGD was longer than that of endomorphin-1 in both the brain homogenate (259.73 ± 41.01 min vs. 23.54 ± 0.25 min) and plasma (262.61 ± 25.20 min vs. 6.09 ± 0.31 min), which explains the longer duration of HAGD antinociception in mice in this study.

In other preclinical pain models, HAGD maintained potent analgesia, similar to morphine, but was stronger than morphine in terms of analgesia intensity (41.11 ± 6.40 vs. 67.36 ± 3.93, *p* < 0.05) against chemical stimulation pain in phase I of the formalin test. Previous studies have shown that endomorphins are better than morphine in a cold-water allodynia test in rats with sciatic nerve injury [18]. Pasquinucci et al. and Vicario et al. also reported that the simultaneous activation of MOR and DOR exhibited potent antinociceptive properties in inflammatory and neuropathic pain modulation [19,20,21]. Similarly, the simultaneous activation of MORs and DORs by HAGD in our study appears to induce potent analgesic effects.

Tolerance and addiction must be considered when developing opioids. Gendron et al. [22] reviewed the functional crosstalk between MORs and DORs, which is related to the development of tolerance. In our study, HAGD exhibited non-tolerance antinociception for eight consecutive days, while morphine produced antinociceptive tolerance in the tail-flick test. It has been reported that the mechanisms of morphine tolerance are multifactorial and complicated, including opioid receptor desensitization and downregulation, alterations in the glutamate receptor, glial activation, and release of inflammatory factors [23]. Therefore, the non-tolerance antinociception of HAGD may be associated with the activation of peripheral MOR and DOR.

The addiction properties of opioid analgesics are partly associated with the activation of dopaminergic reward circuits, which are mediated via MORs in the nucleus accumbens core and DORs in the nucleus accumbens shell, hippocampus, amygdala, striatum, and other basal ganglia structures [24,25]. In the conditioned place preference experiment, mice exhibited a significant conditioned place preference when injected with morphine rather than with HAGD. Regarding the withdrawal response, mice displayed a significant physical dependence on both HAGD and morphine, which is contrary to the above conclusion that HAGD cannot penetrate the BBB. This result is similar to peripheral DN-9 administration (s.c.) in mice. DN-9 produced peripherally restricted antinociception and psychological dependence in the conditioned place preference test but not physical dependence in the withdrawal response [16]. We conjecture that the BBB may have low permeability for HAGD and the doses penetrating the BBB can elicit an addictive response but are not sufficient for antinociception. Activation of dopaminergic reward circuits not only partially contributes to opioid addiction, but also produces an acute hyperlocomotive response in the acute hyperlocomotion test [26,27], which is in accordance with the acute hyperlocomotive response observed in mice injected with morphine but not HAGD. This result may be attributed to the low BBB permeability of HAGD. HAGD had limited effects on GIT, whereas morphine inhibited GIT to a greater extent (44.83 ± 4.14 vs. 24.11 ± 3.79, *p* < 0.05). Furthermore, HAGD and morphine did not have remarkable effects on mice in the rotating rod test. These results underscore the safety of HAGD.

Overall, HAGD did not have significant negative effects related to tolerance, psychological dependence, acute hyperlocomotive responses, or motor coordination at potent analgesic doses in mice. Studies have shown that simultaneously targeting multiple opioid receptors may reduce side effects [28,29], and the simultaneous targeting of MOR and DOR in this study resulted in a safer, but equipotent, analgesic response to that of morphine.

Reduced sperm motility (asthenozoospermia) [30] is a common abnormality in people with opiate addiction. In 2006, the presence of functional MORs, DORs, and KORs in human sperm membranes was reported for the first time [7]. The effects of exogenous and endogenous opioids on human sperm motility via the three classical opioid receptors have been reported [6,8,14,31]. In this study, different concentrations of HAGD (10, 1, 0.1, 0.01, 0.001, and 0.0001 µmol/L) did not have a significant negative effect on human sperm motility within the duration of the experiment. Consistent with previous research [7], 0.1 µmol/L (1 × 10^−7^ mol/L) morphine significantly reduced human sperm motility after 3.5 h. We also found that there was also a significant inhibition after 3.5 h when the morphine concentration was 0.01 µmol/L (1 × 10^−8^ mol/L). This finding may be closely related to the ratio of MORs/DORs activated by different morphine concentrations. Overall, HAGD did not reduce human sperm motility in vitro.

In summary, this study illustrates that HAGD, as a multi-target MOR/DOR agonist, produces a potent antinociceptive effect in various preclinical pain models. The side effects of HAGD on tolerance development, rewarding effects, constipation, and sperm motility were significantly reduced compared with those of morphine. HAGD may be a promising compound and will be further optimized to promote the transformation of basic medicine into clinical medicine in the future.

## 4. Materials and Methods

### 4.1. Drugs and Chemicals

Morphine hydrochloride was produced by the Shenyang First Pharmaceutical Factory (Shenyang, China). Naloxone, naloxone methiodide, β-funaltrexamine hydrochloride, nor-binaltorphimine dihydrochloride, and naltrindole were purchased from Sigma-Aldrich (St. Louis, MO, USA). All amino acids were purchased from GL Biochem Ltd. (Shanghai, China). All organic chemicals were purchased from J&K Scientific Ltd. (Beijing, China). Rink amide 4-methylbenzhydrylamine (MBHA) resin was purchased from Nankai Hecheng Science & Technology Co., Ltd. (Tianjin, China). The G-MOPS^TM^ PLUS was obtained from Vitrolife Sweden AB (Goteborg, Sweden). Drugs for mice were dissolved in physiological saline, and those for sperm were dissolved in G-MOPS^TM^ PLUS. All drugs were stored at −20 °C. G-MOPS ^TM^ PLUS was stored at 4 °C.

### 4.2. Peptide Synthesis

HAGD was prepared by *fluorenylmethoxycarbonyl (Fmoc)* solid-phase synthesis on Rink amide MBHA resin according to previous studies [32,33]. The compounds 2-(1H-benzotriazole-1-yl)-1,1,3,3-tetramethyluronium hexafluorophosphate (HBTU), 1-hydroxybenzotriazole (HOBT), and N,N′-diisopropylethylamine (DIEA) were used as amino acid coupling agents. The protected peptide-resin was cleaved with trifluoroacetic acid (TFA)/triisopropylsilane/H_2_O (95:2.5:2.5). The crude peptide was extracted with 10% acetic acid solution and purified using semipreparative reversed-phase high-pressure liquid chromatography (RP-HPLC). A white powder was obtained after lyophilization. The molecular weight was measured using electrospray ionization time-of-flight mass spectrometry (ESI-Q-TOF maXis-4G, Bruker Daltonics, Germany) to identify the peptide structure. The purity of the peptide was determined to be 95% via analytical RP-HPLC; this value was used in the subsequent analyses.

### 4.3. Administration

Naloxone (1 mg/kg equivalent to 2.5 μmol/kg, s.c.) or naloxone methiodide (1 mg/kg equivalent to 2.13 μmol/kg, s.c.; 5 nmol, intracerebroventricular (i.c.v.)) was injected 10 min prior to the subcutaneous injection of HAGD (10 mg/kg equivalent to 21.98 μmol/kg) in the tail-flick test in this study [34]. The selective antagonists, detailed in Table 2, for MOR, DOR, and kappa opioid receptors (KOR), i.e., β-funaltrexamine hydrochloride (1 mg/kg equivalent to 1.72 μmol/kg), naltrindole (1 mg/kg equivalent to 2.07 μmol/kg), and nor-binaltorphimine dihydrochloride (1 mg/kg equivalent to 1.19 μmol/kg), respectively, were subcutaneously (s.c.) injected 4 h, 10 min, or 30 min, respectively, prior to the subcutaneous injection of HAGD (10 mg/kg equivalent to 21.98 μmol/kg) in the tail-flick test [34]. The doses of opioid receptor antagonists were based on the previous studies [16,17].

### 4.4. Animals

Briefly, 6–8-week-old male Kunming mice (18–22 g) were provided by the Experimental Animal Center of Lanzhou University. The mice were housed in a standard animal room maintained at 22 ± 1 °C with a 12/12-h dark/light cycle. Food and water were provided ad libitum. Animal experimental protocols were carried out in accordance with the European Community guidelines (2010/63/EU). Each group included at least five mice, with great efforts to minimize the number of mice used. A total of 358 mice were used in this study. Before the experiments, the mice were allowed to acclimatize to the environment for 30 min. The study was blinded to the treatment assignment and data analysis.

### 4.5. Calcium Mobilization Assays

CHO (CHO_hMOP_, CHO_hDOP_, and CHO_hKOP_) cell lines (GenScript) were seeded in black and clear-bottom 96-well plates at a density of 30,000 cells/well and incubated at 37 °C in 5% CO_2_ overnight [40,41]. The old medium was discarded, and 2.5 mM probenecid (Abmole Bioscience), 1.35 μM calcium-sensitive fluorescent dye Fluo-4 AM (Enzo Life), and 0.1% Pluronic F-127 (Sigma-Aldrich) Hank’s balanced salt solution (HBSS) were added per well for 30 min at 37 °C. The dye solution was replaced with 75 µL of HBSS with 2.5 mM probenecid for 15 min. After adding 25 µL of HAGD at different concentrations to FlexStation 3 (Molecular Devices) at 485 nm excitation wavelength and 525 nm emission wavelength, fluorescence changes were recorded. All experiments were performed in duplicate and repeated at least three times.

### 4.6. The Tail-Flick Test

The radiant heat tail-flick test has been widely used as an acute pain model to examine the effects of analgesics [16]. The mice were gently stabilized by hand, and the dorsal surface of the tail, 3 cm from the distal end, was placed on a radiant heat source. The cutoff time was set to 10 s to avoid tissue damage. Radiant heat intensity was adapted to engender a basal latency within 3–5 s in naive mice. Tail-flick latency was defined as the time when the mouse flicked its tail away from the radiant heat source. The data of antinociceptive effects for each animal after treatment were calculated as a percentage of the maximal possible effect (MPE%): MPE (%) = 100 × [(postdrug response − baseline response)/(cutoff response − baseline response)]. The MPE% from each animal was converted to an area under the curve (AUC). The AUC value was calculated over a 90-min period.

### 4.7. Carrageenan-Induced Inflammatory Pain

Carrageenan has been used in the inflammatory pain model [42]. The mice were placed in a clear plastic chamber on the glass surface of radiant heat equipment (PL-200, Chengdu Technology & Market Co., Ltd., Chengdu, China). Radiant heat intensity was adapted to engender a basal latency within 10–20 s in naive mice. The cutoff time was set at 25 s to avoid tissue damage. On the first day, the paw withdrawal threshold was tested, and the right hind paw intraplantar injection with 20 μL of 2% λ-carrageenan was performed. After 24 h, thermal hyperalgesia was measured. Subsequently, the paw withdrawal threshold was measured three times at approximately 2-min intervals at every point after administration. Percent reversal of hyperalgesia for each animal after treatment was defined as: MPE (%) = 100 × [(postdose threshold − predose threshold)/(baseline threshold − predose threshold)]. Similarly, the MPE% was converted to the AUC and the latter was calculated from 0 to 90 min.

### 4.8. Acetic Acid-Induced Writhing Test

An acetic acid-induced writhing test was performed to investigate visceral pain [43]. The increase in intensity of drug analgesia would be accompanied by a decrease in the writhing number. The mice were acclimatized in a transparent acrylic chamber (20 × 20 × 30 cm) for 15–20 min. First, the mice were subcutaneously injected with drugs. Five minutes later, they were intraperitoneally injected with 0.6% acetic acid solution. Within 5–15 min after injecting acetic acid, the number of writhes was recorded. Writhe was defined as a contraction of the abdominal muscles and stretching of the hind limbs.

### 4.9. Formalin Test

The formalin test was conducted to explore a characteristic biphasic pain response [44]. The better the analgesic effect in mice, the shorter the average licking/biting/shaking paw time. The mice were acclimated in a transparent plexiglass chamber (20 × 20 × 30 cm) for 15–20 min. First, the mice received a subcutaneous injection of drugs. Five minutes later, 20 μL of 5% formalin was administered by intraplantar injection to the right hind paw. The mice were immediately placed back in the chamber. Simultaneously, the time spent licking, biting, and shaking the injected paw was recorded within 0–5 and 15–30 min.

### 4.10. Antinociceptive Tolerance

A tolerance test was performed in the tail-flick test [16]. The mice were subcutaneously injected with saline, HAGD (10 mg/kg equivalent to 21.98 μmol/kg), or morphine (10 mg/kg equivalent to 31.07 μmol/kg) once a day for 8 days. The tail-flick latency was determined 15 min after HAGD injection and 30 min after morphine injection when antinociception peaks. A reduction in tail-flick latency over an 8-day time course was suggestive of a tolerance effect.

### 4.11. Naloxone-Induced Withdrawal Response

Jumping is the main behavioral symptom of naloxone-induced withdrawal response. An increase in the jump number after naloxone injection was considered indicative of withdrawal response. The mice were subcutaneously injected with seven increasing doses of HAGD or morphine (20, 40, 60, 80, 100, 100, and 100 mg/kg) every 8 h [45]. Two hours after the last challenge, the mice were subcutaneously administered naloxone (10 mg/kg equivalent to 25 μmol/kg) and placed in a separate cylinder with a height of 32 cm and a diameter of 12 cm. The jump number was recorded for 30 min [17].

### 4.12. Conditioned Place Preference

The conditioned place preference experiment is a classical model for evaluating dependence [42]. The device consisted of three compartments. Two large compartments (20 × 20 × 20 cm) were connected by a narrower compartment (5 × 20 × 20 cm). The large compartments were visually and tactually distinct (white walls with rough floors and black walls with smooth floors). Before the unbiased experiment, the mice were allowed to move freely for 15 min to accommodate the apparatus. On the first day, the time the mice spent in each compartment was documented within 15 min. Over the next 3 days, the mice were subcutaneously administered drugs and were limited to one of the two large compartments. Approximately 6 h later, the mice were subcutaneously injected with saline and confined to the opposite compartment. On the fifth day, the mice moved freely for 15 min, and the time spent in the drug-associated preference compartment was recorded. The conditioned place preference score was calculated as: score = time spent in the drug-associated preference compartment on the fifth day − time spent in the same compartment on the first day. An increase in the conditioned place preference score indicated that mice had a conditional place preference following drug administration.

### 4.13. Acute Hyperlocomotion Test

The open-field test was performed to assess locomotor activity [46]. An increase in the total distance moved by mice after analgesic drug administration indicated that analgesics can induce acute hyperlocomotion. The apparatus consisted of an uncovered black plexiglass arena (50 × 50 × 40 cm) and a video tracking system (PMT-100, Chengdu Technology & Market Co., Ltd., Chengdu, China). Initially, the mice were placed in the center of the arena and allowed to explore freely for 30 min. The mice were then subcutaneously injected with saline, HAGD, or morphine, and locomotor activity was monitored for another 150 min. The experimental arena was wiped with 75% ethanol to eliminate the scent.

### 4.14. Gastrointestinal Transit Test

The mice were fasted for 16 h with free access to water [16,47]. Drugs were subcutaneously injected 15 min before the oral administration of a charcoal meal (an aqueous suspension of 5% charcoal and 10% gum arable). Thirty minutes after the oral administration, the animals were sacrificed. The farthest traveled distance by the charcoal meal (L_1_) and total length of the small intestine (L_2_) were measured. The GIT inhibition by analgesics was represented by GIT% = (L_1_/L_2_) × 100.

### 4.15. Rotarod Test

Motor coordination and equilibrium were determined using a rotarod apparatus (ZB-200, Chengdu Technology & Market Co., Ltd., Chengdu, China) [17,46]. The mice were trained on a rotating rod at 16 rpm thrice daily for 2 days. Animals that remained on the rod for at least 180 s were used. The cutoff time was 300 s. On the third day, the mice received subcutaneous administration of drugs. The latency to fall off the rod was recorded 15, 30, 45, 60, and 90 min after administration. If the motor coordination of mice is impaired after administration of analgesics, the latency is reduced.

### 4.16. Sperm Motility

#### 4.16.1. Sperm Preparation

Freshly ejaculated semen was obtained by masturbation after abstinence (3–7 d). All donors (18–40 years old) had normal sperm parameters according to the World Health Organization [48] and provided informed consent. The experimental protocols were approved by the Ethics Committee of the First Hospital of Lanzhou University (LDYYLL2019-48, 2020–2023). The semen ejaculated into sterile containers was liquefied at 37 °C for 30 min. The samples were processed using the swim-up technique [6,7]. One 15-mL centrifuge tube contained fresh semen (2 mL) and G-MOPS**^TM^** PLUS medium (2 mL). After approximately 45 min of incubation at 37 °C, most of the upper medium layer was transferred to a new centrifuge tube and centrifuged at 1500 rpm for 4 min. Subsequently, the sperm sediment was resuspended at 20 × 10^6^ cells/mL using G-MOPS**^TM^** PLUS. A spermatozoa suspension with at least 60% forward motility was used.

#### 4.16.2. Incubation Medium

The spermatozoa suspension was divided into 20-μL aliquots. The control aliquot was treated with G-MOPS**^TM^** PLUS, and the other aliquots were incubated with different doses of HAGD (180 μL) or morphine (180 μL) for 4 h at 37 °C. All media were prepared on the day of use and maintained at 37 °C. All incubations were performed at 37 °C.

#### 4.16.3. Motility Analysis

Motility analysis was carried out using an automatic sperm analysis system (Beijing Suiplus Software Co., Ltd., Beijing, China) at 0, 0.5, 1, 3.5, and 4 h after drug addition to the medium. To measure sperm concentration and motility, a wet preparation was performed. A minimum of 200 sperms from at least five different fields were analyzed. The motility of each spermatozoon was graded according to the World Health Organization [48], with progressive motility (PR): spermatozoa moving actively, either linearly or in a large circle, regardless of speed; non-progressive motility: all other patterns of motility with an absence of progression; and immotility: no movement. Alterations in the PR reflected the effects of analgesics on sperm motility in vitro.

### 4.17. Statistical Analysis

All data are presented as the means ± standard error of the mean (S.E.M.). Data were analyzed via one-way analysis of variance (ANOVA) followed by Dunnett’s or Bonferroni’s post hoc test, as specified. The dose and time responses were analyzed using two-way ANOVA. The effective dose at 50% of maximum response (ED_50_) in tail-flick test, the effective concentration at 50% of the maximum response (EC_50_), and 95% confidence interval (CI) in the calcium mobilization assays were calculated using GraphPad Prism 7. Statistical significance was set at *p* < 0.05.

## Figures and Tables

**Figure 1 molecules-28-00427-f001:**
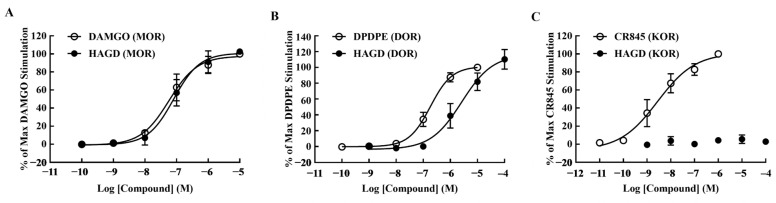
Agonistic activity of HAGD in calcium mobilization assays. Drug dosage was in mol/L in (**A**) CHO_hMOP_, (**B**) CHO_hDOP_, and (**C**) CHO_hKOP_ cells. CHO_hMOP_, CHO_hDOP_, and CHO_hKOP_: Chinese hamster ovary cells expressing human mu, delta, and kappa opioid receptors, respectively. MOR, mu opioid receptor; DOR, delta opioid receptor; KOR, kappa opioid receptor; M, mol/L.

**Figure 2 molecules-28-00427-f002:**
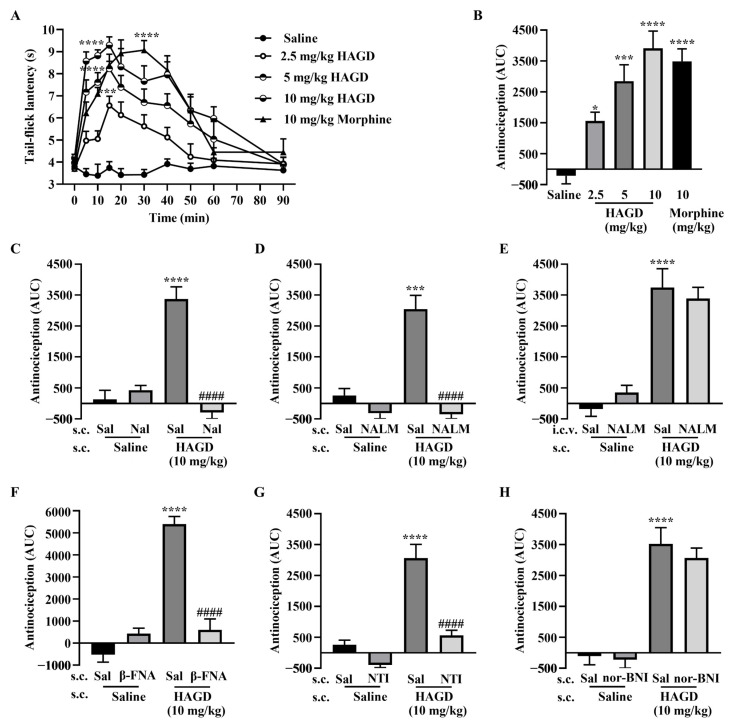
Antinociceptive effects of subcutaneous (s.c.) HAGD in the tail-flick test. (**A**) Antinociceptive dose- and time-response curve for HAGD and morphine; *n* = 6, 5, 5, 6, 6 mice; *** *p* < 0.001 and **** *p* < 0.0001 indicate significant differences compared with the saline group, according to two-way analysis of variance (ANOVA), followed by Bonferroni’s post hoc test. (**B**) The area under the curve (AUC) values of the percentage of the maximal possible effect (MPE%) of HAGD and morphine during the observed period; *n* = 6, 5, 5, 6, 6 mice; * *p* < 0.05, *** *p* < 0.001 and **** *p* < 0.0001 indicate significant differences compared with the saline group, according to one-way ANOVA, followed by Dunnett’s post hoc test. (**C**) Pretreatment with s.c. naloxone (Nal, 1 mg/kg equivalent to 2.5 μmol/kg, 10 min); *n* = 5, 5, 5, 6 mice; (**D**) Pretreatment with s.c. naloxone methiodide (NALM, 1 mg/kg equivalent to 2.13 μmol/kg, 10 min); *n* = 5, 5, 5, 6 mice; (**E**) Pretreatment with intracerebroventricular (i.c.v.) NALM (5 nmol, 10 min); *n* = 5, 5, 5, 6 mice; (**F**) s.c. β-funaltrexamine hydrochloride (β-FNA, 1 mg/kg equivalent to 1.72 μmol/kg, 4 h) blocked the antinociception of HAGD; *n* = 5, 6, 5, 6 mice; (**G**) s.c. naltrindole (NTI, 1 mg/kg equivalent to 2.07 μmol/kg, 10 min) blocked the antinociception of HAGD; *n* = 5, 5, 5, 6 mice; (**H**) s.c. nor-binaltorphimine dihydrochloride (nor-BNI, 1 mg/kg equivalent to 1.19 μmol/kg, 30 min) had no effect on the antinociception of HAGD; *n* = 5, 5, 5, 5 mice; *** *p* < 0.001 and **** *p* < 0.0001 indicate significant differences compared with the saline + saline group, #### *p* < 0.0001 indicates significant differences compared with the saline + HAGD group, according to one-way ANOVA, followed by Bonferroni’s post hoc test. Sal, saline.

**Figure 3 molecules-28-00427-f003:**
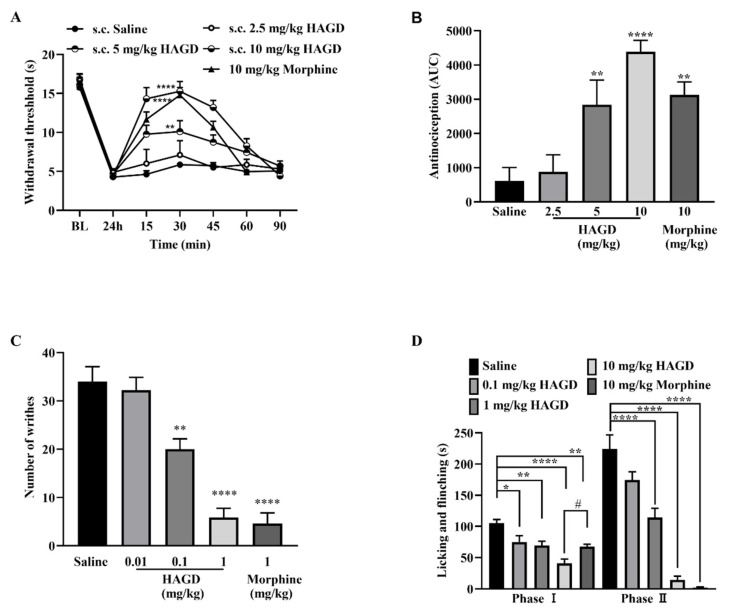
Antinociceptive effects of subcutaneous (s.c.) HAGD. (**A**) Antinociceptive dose- and time-response curve for HAGD and morphine in the carrageenan-induced inflammatory pain, *n* = 9, 8, 9, 8, 9 mice; ** *p* < 0.01 and **** *p* < 0.0001 indicate significant differences compared with the saline group, according to two-way analysis of variance (ANOVA), followed by Bonferroni’s post hoc test; (**B**) The area under the curve (AUC) values of the percentage of the maximal possible effect (MPE%) of HAGD and morphine; ** *p* < 0.01 and **** *p* < 0.0001 indicate significant differences compared with the saline group, according to one-way ANOVA, followed by Dunnett’s post hoc test. (**C**) Dose-dependent histogram of HAGD and morphine in the acetic acid-induced writhing test, *n* = 5, 6, 6, 5, 5 mice; ** *p* < 0.01 and **** *p* < 0.0001 indicate significant differences compared with the saline group, according to one-way ANOVA, followed by Dunnett’s post hoc test. (**D**) Dose-dependent histogram of HAGD and morphine in the formalin test, *n* = 6, 5, 7, 7, 6 mice; * *p* < 0.05, ** *p* < 0.01, and **** *p* < 0.0001 indicate significant differences compared with the saline group, # *p* < 0.05 indicated significant differences compared with the morphine group, according to one-way ANOVA, followed by Dunnett’s post hoc test.

**Figure 4 molecules-28-00427-f004:**
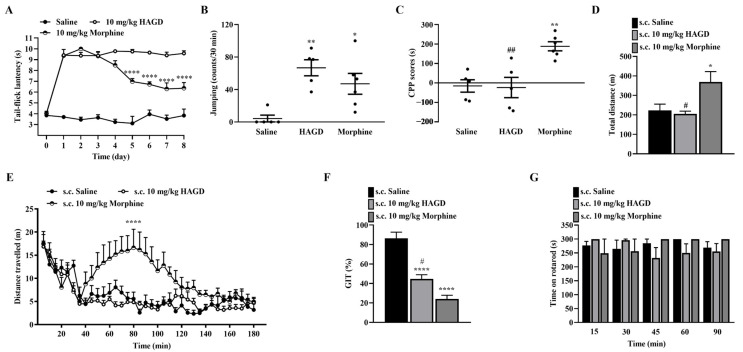
Evaluation of side effects of subcutaneous (s.c.) HAGD (10 mg/kg equivalent to 21.98 μmol/kg) and morphine (10 mg/kg equivalent to 31.07 μmol/kg). (**A**) Effects of HAGD on tolerance; *n* = 5, 7, 5 mice. **** *p* < 0.0001 indicates significant differences compared with the nociceptive latency on day 1, according to two-way analysis of variance (ANOVA), followed by Bonferroni’s post hoc test. (**B**) Effects of HAGD on withdrawal response; *n* = 5, 6, 6 mice. (**C**) Effects of HAGD on conditioned place preference (CPP); *n* = 5, 5, 6 mice. (**D**,**E**) Effects of HAGD on acute hyperlocomotion test; *n* = 6, 6, 6 mice. (**F**) Effects of HAGD on gastrointestinal transit (GIT); *n* = 6, 6, 6 mice. (**G**) Effects of HAGD on rotarod test; *n* = 6, 6, 5 mice. * *p* < 0.05, ** *p* < 0.01 and **** *p* < 0.0001 indicate significant differences compared with the saline group, # *p* < 0.05 and ## *p* < 0.01 indicate significant differences compared with the morphine group, according to one-way ANOVA, followed by Bonferroni’s post hoc test.

**Figure 5 molecules-28-00427-f005:**
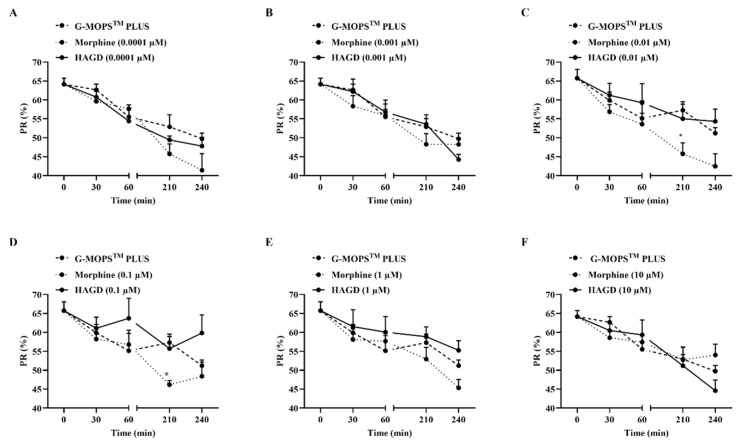
Evaluation of the effects of HAGD on sperm motility (*n* = 5 for each drug concentration). (**A**) To evaluate the effect of 0.0001 µM of HAGD on sperm motility. (**B**) To evaluate the effect of 0.001 µM of HAGD on sperm motility. (**C**) To evaluate the effect of 0.01 µM of HAGD on sperm motility. (**D**) To evaluate the effect of 0.1 µM of HAGD on sperm motility. (**E**) To evaluate the effect of 1 µM of HAGD on sperm motility. (**F**) To evaluate the effect of 10 µM of HAGD on sperm motility. * *p* < 0.05 indicates significant differences compared with the G-MOPS**^TM^** PLUS group, according to two-way analysis of variance, followed by Bonferroni’s post hoc test. Drug dosage was in µmol/L for each sperm sample. PR, progressive motility.

**Table 1 molecules-28-00427-t001:** The agonist efficacy of HAGD in calcium mobilization assays.

Compound	pEC_50_ (95% CI)	*E*_max_ ± S.E.M.	EC_50_ (nM)	R^2^
DAMGO (MOR)	7.24 (7.02–7.44)	100.00 ± 0.00	58.01	0.98
HAGD (MOR)	7.09 (6.85–7.29)	102.56 ± 1.58	81.69	0.97
DPDPE (DOR)	6.74 (6.61–6.85)	100.00 ± 0.00	183.1	0.99
HAGD (DOR)	5.58 (5.06–5.9)	110.36 ± 7.21	2628	0.97
CR845 (KOR)	8.56 (8.18–8.89)	100.00 ± 0.00	2.79	0.95
HAGD (KOR)	^(-)^	2.95 ± 0.92	^(-)^	^(-)^

^(-)^ The data cannot be determined. pEC_50_, the negative logarithm of the molar concentration required to stimulate 50% of the opioid receptors present; CI, confidence interval; *E*_max_, the maximum stimulation response; S.E.M., standard error of the mean; EC_50_, the effective concentration at 50% of the maximum stimulation response; nM, nmol/L; R^2^, goodness of fit of the non-linear regression line; MOR, mu opioid receptor; DOR, delta opioid receptor; KOR, kappa opioid receptor.

**Table 2 molecules-28-00427-t002:** The actions of agonists and antagonists at opioid receptors.

Compound	Molecular Weight	*K_i_*^(MOR)^ (nM)	*K_i_*^(DOR^) (nM)	*K_i_*^(KOR)^ (nM)	Agonist	Antagonist	Dose ^a^ (mg/kg, μmol/kg)
morphine [35]	321.85	1.0	125.9	50.1	MOR DOR KOR		10, 31.07
HAGD [14]	455	2.1	3800		MOR DOR		10, 21.98
naloxone [36]	399.87	1.9	3.1	32.6		MOR DOR KOR	1, 2.5
naloxone methiodide [36]	469.31	28.9	1010	203.5		MOR DOR KOR	1, 2.13
β-funaltrexamine [37,38]	581.01	0.3				MOR ^b^	1, 1.72
naltrindole [37]	482.48		0.02			DOR	1, 2.07
nor-binaltorphimine [39]	842.81			0.00025		KOR	1, 1.19

The inhibition constant (*Ki*) represented the agonist or antagonist affinity. ^a^ Drug dosage was in mg/kg (or µmol/kg) for each animal (subcutaneously). ^b^ β-funaltrexamine binds covalently to MOR. MOR, mu opioid receptor; DOR, delta opioid receptor; KOR, kappa opioid receptor. nM, nmol/L.

## Data Availability

The data presented in this study are available on request from the corresponding author.

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
