# Peer review of "A Novel Multi-Target Mu/Delta Opioid Receptor Agonist, HAGD, Produced Potent Peripheral Antinociception with Limited Side Effects in Mice and Minimal Impact on Human Sperm Motility In Vitro"

_molecules, 2023, doi:10.3390/molecules28010427_

Round 1

Reviewer 1 Report

The authors of  A novel multi-target mu/delta opioid receptor agonist, HAGD,  produced potent peripheral antinociception with limited side effects in mice and minimal impact on human sperm motility in vitro nicely covered the develop a novel opioid analgesic, HAGD (H-Tyr-D-AIa-GIy-Phe-NH2), with limited side effects.

In which they illustrate that HAGD, as a multi-target µ/δ opioid receptor agonist, produces a potent antinociceptive effect in various preclinical pain models. I agree with the author that HAGD can be a promising compound and need to do further clinical trials so that it can have a good impact on society.   I strongly recommend publishing the work in a given format. 

Author Response

Responses to Reviewer 1

Comments 1: The authors of A novel multi-target mu/delta opioid receptor agonist, HAGD, produced potent peripheral antinociception with limited side effects in mice and minimal impact on human sperm motility in vitro nicely covered the develop a novel opioid analgesic, HAGD (H-Tyr-D-AIa-GIy-Phe-NH2), with limited side effects.

In which they illustrate that HAGD, as a multi-target µ/δ opioid receptor agonist, produces a potent antinociceptive effect in various preclinical pain models. I agree with the author that HAGD can be a promising compound and need to do further clinical trials so that it can have a good impact on society.   I strongly recommend publishing the work in a given format. 

Response 1: Thank you for your careful review and approval.

Reviewer 2 Report

Manuscript: molecules-2104824

Title: A novel multi-target mu/delta opioid receptor agonist, HAGD, produced potent peripheral antinociception with limited side effects in mice and minimal impact on human sperm motility in vitro

Authors: Li et al.

The manuscript presents effects of a novel opioid analgesic, HAGD (H-Tyr-D-AIa-GIy-Phe-NH2) in mouse models as well as in vitro studies on CHOhMOP, CHOhDOP, and CHOhKOP cell lines and human sperm. The presented observations suggests that HAGD could be useful in the fight against pain in clinical settings. The manuscript is potentially interesting. It should be noted, however, that the manuscript contains some deficiencies and errors. They should be corrected before the manuscript could be accepted for publication. 

List of deficiencies and errors: 

  1. Abstract. Description of methods and results is chaotic. The authors stated that “HAGD antinociception was assessed in preclinical mouse pain models. CHOhMOP, CHOhDOP, and CHOhKOP cell lines were used for calcium mobilization assays”. After that, the authors probably go back to the mouse (no exact data) on which they studied “side effects, including tolerance, withdrawal response, conditioned place preference, acute hyperlocomotion, gastrointestinal transit, and rotarod tests.” The authors should be consistent in presenting the models used, without excessive detail, which will be presented when describing the results. For this purpose, the authors can use the form present in the introduction to the review “The research was curried out in mouse models as well as in vitro studies on CHOhMOP, CHOhDOP, and CHOhKOP cell lines and human sperm.” On the other hand, the results of all tests should be presented in detail.
  2. All abbreviations should be presented in their full name at the point where they appear for the first time, starting from the abstract. Full names of abbreviation should be repeated in, the body of the manuscript at the place of the first use, as well as in Figure legends. Figures should be understandable without having to read the text of the manuscript.
  3. Materials and methods. Authors should report the total number animals used in the study. Has the research been approved by any ethical committee? of If so, the authors should provide the name of the etic committee, the number of the research consent, the date of its granting and the date of expiry of the consent.
  4. Materials and methods. 2.1. Drugs and administration. The authors should provide more details on the synthesis of HSGD, as well as the methods of verifying its structure and the degree of purity of obtained product. In addition, the authors should state what was the purity of HAGD used in the study.
  5. Materials and methods. 2.1. Drugs and administration. In separate table, the authors should present all opioid receptors agonists and antagonist of used in the study. Their molecular weights and affinity for opioid receptors should also be presented. In addition, in the case of antagonists, their specificity and the nature of binding to the receptor (reversible competitive or irreversible covalent) should be provided. This information in the table should be accompanied with appropriate references.
  6. Materials and methods, Figure legends, Results and Discussion. The doses of agonists and antagonists of opioid receptors should be expressed on the basis of their molecular weight. In addition to doses expressed in mg/kg, authors should present doses in millimoles or micromoles per kilogram, in the sections of manuscript mentioned above.
  7. Materials and methods. 2.1. Drugs and administration, Lines 74-76. The authors stated that “Naloxone (1 mg/kg, s.c.) or naloxone methiodide [1 mg/kg, s.c.; 5 nmol, intracerebroventricular (i.c.v.)] was injected 10 min prior to the subcutaneous injection of HAGD (10 76 mg/kg).” In methods, results and figures, authors should clearly indicate in which tests naloxone was used and where naloxone methiodide. Molecular weight of naloxone is 327.4 D, whereas molecular weight of naloxone methiodide is 469,3 D. It indicates that on molecular basis dose 10 mg/kg of naloxone is by 43% higher dose then 10mg/kg of naloxone methiodide.
  8. Materials and methods. 2.1. Drugs and administration (lines 71-77) and Figure 2. Some results are strange. The doses of non-selective and selective opioid receptors antagonists were more than 10-fold lower than dose HAGD. However, data presented in Fig. 2 indicate those low doses of naloxone, naloxone methiodide, β-funaltrexamine hydrochloride and naltrindole. These effects would ultimately be acceptable with an irreversible covalent opioid antagonist. However, how is this possible with reversible competitive antagonist? This issue should be addressed in the Results and Discussion, and the authors should try to provide some plausibly reason for such effect. Another possibility is to repeat this part of study.
  9. Materials and methods, sections 2.4 to 2.14. The authors should present how interpretate those tests.

Author Response

Responses to Reviewer 2

Reviewer 3 Report

The manuscript entitled "A novel multi-target mu/delta opioid receptor agonist, HAGD, produced potent peripheral antinociception with limited side effects in mice and minimal impact on human sperm motility in vitro", from the authors Fangfang Li, Feng Yue, Wei Zhang, Biao Xu, Yiqing Wang and Xuehong Zhang.

The manuscript is very good.

I consider that the manuscript should be published in the journal Molecules.

Author Response

Responses to Reviewer 3

Comments 1: The manuscript entitled "A novel multi-target mu/delta opioid receptor agonist, HAGD, produced potent peripheral antinociception with limited side effects in mice and minimal impact on human sperm motility in vitro", from the authors Fangfang Li, Feng Yue, Wei Zhang, Biao Xu, Yiqing Wang and Xuehong Zhang.

The manuscript is very good.

I consider that the manuscript should be published in the journal Molecules.

Response 1: Thank you for your careful review and approval.

Round 2

Reviewer 2 Report

Manuscript ID: molecules-2104824

Title: A novel multi-target mu/delta opioid receptor agonist, HAGD, 2 produced potent peripheral antinociception with limited side effects in mice and minimal impact on human sperm motility in vitro

Authors: Fangfang Li et al.

The new version of the manuscript shows a significant improvement, but it should be noted that there are still some irregularities.

  1. Abstract 20-22. “The sentence “The analgesic tolerance, rewarding effects (i.e., conditioned place preference and acute hyperlocomotion), and gastrointestinal inhibition of HAGD were significantly reduced compared with those of morphine”. This sentence is strange and unclear, and should be corrected. What does “gastrointestinal inhibition of HAGD” mean? What kind of “gastrointestinal inhibition” evoked by HAGD did the authors mean?
  2. Previous comment 6. “Materials and methods, Figure legends, Results and Discussion. The doses of agonists and antagonists of opioid receptors should be expressed on the basis of their molecular weight”. The authors added doses in micromoles, however, the form used is unclear. For example, the statement in the line 89, “Naloxone (1 mg/kg or 2.5 μmol/kg, s.c.)” may suggest that authors used naloxone in two doses, 1 mg/kg, and 2.5 μmol/kg. For this reason, this statement should be replaced by “Naloxone (1 mg/kg equivalent to 2.5 µmol/kg, s.c.). The same form should be used for the presentation of doses of other compounds used in all parts of the manuscript.
  1. Previous comment 8. Materials and methods. 2.1. Drugs and administration (lines 71-77) and Figure 2. Some results are strange. The doses of non-selective and selective opioid receptors antagonists were more than 10-fold lower than dose HAGD. However, data presented in Fig. 2 indicate those low doses of naloxone, naloxone methiodide, β-funaltrexamine hydrochloride and naltrindole were able completely reverse the effects of HAGD. These effects would ultimately be acceptable with an irreversible covalent opioid antagonist. However, how is this possible with reversible competitive antagonist? This issue should be addressed in the Results and Discussion, and the authors should try to provide some plausibly reason for such effect. Another possibility is to repeat this part of study. In the new version of the Discussion, the authors only stated that opioid receptor antagonists reversed the effects of HAGD, but they should also restated the doses in micromoles/kg of the receptor agonists and antagonists used, the relationship between these doses, as well as their affinity for these receptor, and try to explain how the low doses of competitive antagonists used in the studies were able to revers the effects of opioid receptor agonists given in the doses more than ten times higher than doses of antagonists.

Author Response

Responses to Reviewer 2
